



# CLIGEN Parameter Regionalization for Mainland China

Wenting Wang[1,2], Shuiqing Yin[2], Bofu Yu[3], Shaodong Wang[2]

[1] Zhuhai Branch of State Key Laboratory of Earth Surface Processes and Resource Ecology, Beijing Normal University at Zhuhai, Zhuhai 519087, People's Republic of China

[2] State Key Laboratory of Earth Surface Processes and Resource Ecology, Faculty of Geographical Science, Beijing Normal University, Beijing 100875, People's Republic of China

[3] Australian Rivers Institute, School of Engineering and Built Environment, Griffith University, Nathan, Qld 4111, Australia

*Correspondence to*: Shuiqing Yin (yinshuiqing@bnu.edu.cn)

**Abstract.** Stochastic weather generator CLIGEN can simulate long-term weather sequences as input to WEPP for erosion predictions. Its use, however, has been somewhat restricted by limited observations at high spatial-temporal resolutions. Long-term daily temperature, daily and hourly precipitation data from 2405 stations and daily solar radiation from 130 stations distributed across mainland China were collected to develop the most critical set of site-specific parameter values for CLIGEN. Universal Kriging (UK) with auxiliary covariables, longitude, latitude, elevation, and the mean annual rainfall was used to interpolate parameter values into a 10 km × 10 km grid and parameter accuracy was evaluated based on leave-one-out cross-validation. The results demonstrated that Nash-Sutcliffe efficiency coefficients (NSEs) between UK interpolated and observed parameters were greater than 0.85 for all parameters apart from the standard deviation of solar radiation, skewness coefficient of daily precipitation, and cumulative distribution of relative time to peak intensity, with relatively lower interpolation accuracy (NSE > 0.66). In addition, CLIGEN simulated daily weather sequences using UK-interpolated and observed inputs showed consistent statistics and frequency distributions. The mean absolute discrepancy between the two sequences in the average and standard deviation of the temperature was less than 0.51℃. The mean absolute relative discrepancy for the same statistics for solar radiation, precipitation amount, duration and $I_{30}$ were less than 5%. CLIGEN parameters at the 10 km resolution would meet the minimum WEPP climate requirements throughout in mainland China. The dataset is availability at http://clicia.bnu.edu.cn/data/cligen.html and http://doi.org/10.12275/bnu.clicia.CLIGEN.CN.gridinput.001 (Wang et al., 2020).

**Keywords:** CLIGEN, input parameters, database, China, storm pattern

## 1 Introduction

Weather generators (WGs) are stochastic models that can generate arbitrarily long sequences of weather variables with statistical properties that are similar to observations for a specific location or area (Yin and Chen, 2020). Early WGs were originally developed to provide surrogate climate series for hydrological, soil erosion, and agricultural models when the observed data could not satisfy the application requirements due to missing data, limited record length or spatial coverage (Wilks and Wilby, 1999). Since the 1990s, WGs have received increased attention as a statistical downscaling tool for the assessment of climate change impact (Katz and Parlange, 1996; Maraun et al., 2010). While global climate models (GCMs) /



regional climate models (RCMs) have been used for climate projections, outputs from these models were often too coarse to meet the requirements of earth surface process models in terms of spatial-temporal resolutions and were biased compared

with observations. Statistical downscaling methods, mainly including perfect prognosis (PP), model output statistics (MOS) and WGs, can be used to downscale and bias-correct the output from GCM/RCMs prior to earth surface model applications (Maraun and Widmann, 2018; Yin and Chen, 2020).

CLIGEN is a stochastic WG developed based on the generators used in the EPIC and SWRRB models (Williams et al., 1985; Williams et al., 1984) and was released in 1995 initially accompanied by the process-based soil erosion model

Water Erosion Prediction Project (WEPP) by the United States Department of Agriculture (Nicks et al., 1995). CLIGEN can simulate a series of long-term climate data in daily scale, including maximum and minimum temperatures, precipitation, solar radiation, dew point, wind velocity and direction. In addition, CLIGEN can generate three inter-storm variables in sub-daily scale, including storm duration, time to peak intensity (tp) and the ratio of the peak intensity to the average intensity (ip), from which an unlimited length of high-resolution breakpoint data can be generated (Flanagan et al., 2001; Nicks et al.,

1995; Yu, 2003).

Of the ten CLIGEN-simulated weather elements, seven, namely daily maximum and minimum temperature, daily precipitation, duration, tp, ip, and daily solar radiation, are all that are required for predicting hydrological processes, soil erosion, and bio-production (Arnold et al., 1998; Flanagan et al., 2001; Foster, 2005; Wallis and Griffiths, 1995). These seven climate elements are considered to meet the minimum data requirements for WEPP if modeling wind-induced snow

drift is not needed (Flanagan and Livingston, 1995). As CLIGEN is independent of WEPP, it can be used to provide simulated climate series for other surface process models as well (Flanagan et al., 2014; Yu, 2002).

**Table 1**

Thirteen groups of input parameters related to temperature, solar radiation and precipitation as listed in Table 1 are all parameters needed by CLIGEN to generate the aforementioned seven climate elements. As a site-specific weather

generator, input parameters for CLIGEN can be directly prepared for sites with observed data. CLIGEN was initially released in the United States with a set of 2600 weather station parameter files (Flanagan et al., 2001). Parameters for the daily temperature and daily precipitation were calculated directly based on the observations of temperature and precipitation from each station. Parameters for daily solar radiation and storm pattern were based on 142 weather stations with daily solar radiation and sub-daily rainfall observations first, and then extended to other 2000 more stations using the triangulation

interpolation method (Scheele and Hall, 2000).

Parameter regionalization, which extends model parameter values from stations with observations to areas/regions without observations, is required when the model is going to be used in these areas/regions. Commonly used parameter regionalization methods can be categorized as follows: (1) the parametric transplantation method, where a reference area that is spatially near or has similar climate characteristics to the target area is first selected, then the parameters of the reference

area are extended to the target area (Cheng et al., 2016); (2) spatial interpolation method such as Thiessen polygon, inverse distance weighted, or ordinary Kriging, that interpolate parameter values based on spatial correlations of parameters among



multiple sites (Hutchinson, 1995); (3) parameter transfer as a function of regional properties such as multiple regression, based on correlations between parameters and regional characteristics (Cowpertwait et al., 1996); (4) regionalization considering both the spatial correlation of parameters and the correlation between parameters and regional characteristics,

including external drift Kriging, and universal Kriging, that can be treated as combination methods to take advantage of (2) and (3) (Haberlandt, 1998; Semenov and Brooks, 1999).

Accuracy of parameter regionalization is known to be influenced by several factors. Firstly, regionalization of climate variables with lower or regular spatial variability generally performs better than highly heterogeneous and discontinuous variables. Xu et al. (2018) attempted to regionalize monthly temperature and precipitation in the Kangdian

region of China and noted that the root mean square error (RMSE) of the temperature was less than that of the precipitation. Secondly, for the same climate variable, temporal resolution plays an important role. The climate variable at a monthly or annual scale tends to perform better than variables at a daily or hourly scale because data with finer resolutions possess greater spatial variability. Thirdly, adopted approaches affect the efficiency of regionalization. For example, Wilks (2008) compared and evaluated the interpolation accuracy of four spatial interpolation methods for WGEN parameters and showed

that locally weighted regressions outperformed Thiessen polygons and domain-wide ('global') regressions. The accuracy of interpolation can be improved by adopting auxiliary covariables that are correlated with the regionalized climate variables into the regionalization process (Hengl et al., 2007). For example, elevation is frequently used as an auxiliary covariable and has been found to improve the interpolation of temperature and precipitation (Carrera-Hernández and Gaskin, 2007; Ly et al., 2013; Verworn and Haberlandt, 2011), especially in mountainous regions with complex terrains (Xu et al., 2018).

Several studies have attempted at regionalization of CLIGEN input parameters. Regionalization of CLIGEN input parameters for WEPP have combined the parameter transport and spatial interpolation. When CLIGEN was developed in the U.S. to provide climate input to WEPP, parameter values for 2600 stations were regionalized based on inverse distance weighting (IDW). In the WEPP application, users identify the targeted location, for which daily weather sequences using parameters from the nearest stations will be automatically generated directly or by interpolation from surrounding stations

(up to 20 stations within a distance of one degree of latitude/longitude). The parameter files and the internally installed interpolation in WEPP application has facilitated application of CLIGEN/WEPP in the US. However, the accuracy of regionalized parameters has not been evaluated and the effect on generated weather sequences using the interpolated parameters are largely unknown.

Chen (2008) explored four spatial interpolation methods, inverse distance weighting (IDW), ordinary Kriging (OK),

global polynomial interpolation (GPI), and local polynomial interpolation (LPI), to regionalize the daily temperature and precipitation related input parameters of CLIGEN for 12 stations in the Loess Plateau of China. Paired t-tests show that the temperature and precipitation series generated using interpolated input parameters are not significantly different from those generated using input parameters computed using observations for the 12 sites considered (Chen, 2008). However, solar radiation and storm pattern-related parameters used to generate daily solar radiation and storm characteristics were not

considered in Chen's study (Chen, 2008). Input parameters for simulating the 7 weather variables mentioned above, listed in



Table 1, meet the minimum data requirements for WEPP at a specific site. Without temperature, solar radiation and storm pattern-related parameter values, CLIGEN cannot be used to generate the required weather sequences for WEPP.

The overall aim of this study was to enable widespread use of CLIGEN to generate daily precipitation, temperature, and solar radiation variables anywhere in mainland China and to gain better understanding of the performance of various

spatial interpretation techniques. Specific objectives of this study were to (1) assemble CLIGEN input parameter values for 2405 sites in mainland China based on meteorological observations; (2) evaluate spatial interpolation techniques for regionalizing CLIGEN parameters; (3) produce grid-based CLIGEN temperature, solar radiation and precipitation parameter values at 10 km resolution for mainland China.

## 2 Data and methods

**2.1 Data collection**

Four datasets consisting of daily temperature, daily rainfall, and hourly rainfall from 2405 meteorological stations, and solar radiation data from 130 stations distributed across mainland China were collected (Fig. 1) from the National Meteorological Information Center (NMIC) of the China Meteorological Administration (CMA) and have been quality controlled by NMIC. Data lengths were different for these four datasets (Table 2). Daily temperature and daily rainfall data

were characterized by longer periods of observation for most stations compared with hourly rainfall data, especially for stations located in the northwest arid area and the Qinghai-Tibet plateau where gauges for observing hourly rainfall for some stations were installed very late (Zhao, 1983; Wang and Zuo, 2009). Based on these four data sets, a total of 156 parameter values were calculated for each station. It should be noted that the 12$^{th}$ value of TimePk is equal to 1 by definition and 155 parameters were involved in the calculation and interpolation. The siphon rain gauges used to record hourly rainfall were

stopped in winter to avoid freezing failures; therefore, hourly rainfall was only available for the warm rainy season for some northern and western stations. Nine stations distributed in the North China (Miyun, Zhengzhou, Ha'erbin), Northwest China (Lanzhou, Wulumuqi), the Tibet Plateau (Lasa), and South China (Fuzhou, Changsha, Haikou) were selected to further display the regional differences and monthly variability of input parameters (Fig. 1).

<p style="text-align:center">**Fig. 1.**</p>

<p style="text-align:center">**Table 2.**</p>

**2.2 Site-based input parameters and simulation**

CLIGEN requires 13 groups of input parameters and 12 values for each group to stochastically simulate temperature, solar radiation and precipitation (Table 1). Temperature-related input parameters, TMAX AV, SD TMAX, TMIN AV, and SD TMIN are used to simulate the daily maximum and minimum temperature for each simulated day and to decide whether

the simulated precipitation occurred as snowfall or rainfall (Table 1). These four values can be calculated using daily



maximum and minimum temperature data for each month directly. Solar radiation related inputs SOL.RAD and SD SOL are used to generate daily solar radiation and can be directly obtained from observed daily solar radiation.

The wet-following-wet and wet-following-dry transition probabilities, $P(W|D)$ and $P(W|W)$ are used to determine the occurrence of rainy days with a first-order two-states Markov chain prepared as follows:

$$P(W|W) = \frac{N_{ww}}{N_{wd}+N_{ww}},$$  (1)

$$P(W|D) = \frac{N_{dw}}{N_{dw}+N_{dd}},$$  (2)

in which, $N_{ww}, N_{wd}, N_{dw}, N_{dd}$ represent the number of days in a month that a wet day followed a wet day, a wet day followed a dry day, a dry day followed a wet day, and a dry day followed a dry day, respectively. For each simulated wet day, MEAN P, S DEV P, and SKEW P are used to simulate the daily precipitation amount using a skewness normal distribution. These three parameters can be computed directly from daily precipitation month by month. As CLIGEN assumes there is only one storm occurring on a wet day, daily precipitation amount in CLIGEN are equal to storm precipitation amount.

MX.5P and TimePk are used to simulate inter-storm variables, including storm duration (D, h) and two normalized dimensionless variables, the ratio of peak intensity to average intensity ($i_p$), and the ratio of time to the peak intensity to storm duration ($t_p$) (Nicks et al., 1995; Yu, 2002; Yu, 2003; Zhang and Garbrecht, 2003). MX.5P represents the average maximum 30-min intensity for each month. The maximum 30-min intensity for a wet day is denoted as $I_{30}$. If a month has $n$ wet days, the maximum $I_{30}$ among $n$ wet days can be denoted by $\max I_{30}$; and for a specific month in a data series of $k$ years, the MX.5P is given by:

$$\mathrm{MX.5P} = \frac{1}{k}\sum maxI_{30}.$$  (3)

Theoretically, MX.5P are expected to be prepared using rainfall data with an observed interval $\leq$ 30 min. Considering the limited availability of aforementioned high-resolution rainfall observations, MX.5P was calculated in this study using hourly data in reference to methods developed by Wang et al. (2018b). Rainfall intensity is basically assumed to be ranked from high to low in CLIGEN (Nicks et al., 1995); therefore, the precipitation depth $P_{\Delta t}$ in any given interval $\Delta t$ can be described by:

$$P_{\Delta t} = i_p \int_0^{\Delta t} e^{-t/\tau}dt = \tau i_p\left(1 - e^{-\Delta t/\tau}\right).$$  (4)

For hourly data, the interval $\Delta t = 1$ h, and the maximum 1 h precipitation $P_{1h}$ and maximum 2 h precipitation $P_{2h}$ were known:

$$\frac{P_{1h}}{P_{2h}} = \frac{1-e^{-1/\tau}}{1-e^{-2/\tau}},$$  (5)

where $\tau$ can be solved and then $i_p$ can be readily obtained as,

$$i_p = \frac{P_{1h}}{\tau(1-e^{-\frac{1}{\tau}})}.$$  (6)

Once $\tau$ and $i_p$ are known, the maximum 30-min precipitation $P_{0.5}$ can be determined as,

$$P_{0.5h} = \tau\, i_p(1 - e^{-\frac{1}{2\tau}}).$$  (7)





The maximum 30-min rainfall intensity is given simply as,

$$I_{30min} = 2P_{0.5h}. \tag{8}$$

In reference to Wang et al. (2018b), TimePk can be directly prepared using hourly rainfall data.

There are 12 discrete values of TimePk for each site, describing an empirical cumulative probability distribution of time to peak (Nicks et al., 1995). The observed interval is $\Delta t$ and the storm duration, D, consists of n intervals. If the peak intensity occurs in the $i$th interval, time to peak intensity, $T_p$ is estimated as,

$$T_p = (i - \frac{1}{2})\Delta t, \tag{9}$$

and time to peak as a fraction of duration is,

$$t_p = \frac{T_p}{D} = \frac{(i-0.5)}{n}. \tag{10}$$

If $Ntp(i)$ is the number of wet days from all data records with $t_p \leq i/12$ for $i = 1,2,...12$, then

$$TimePk(i) = \frac{Ntp(i)}{Ntp(12)}. \tag{11}$$

In reference to Wang et al. (2018b), TimePk was prepared directly using hourly data as well as MX.5P.

## 2.3 Spatial interpolation by Kriging

Kriging interpolation is a spatial interpolation method that gives the best linear unbiased prediction of intermediate values, assuming a Gaussian process governed by prior covariance. For a research region with *n* samples at spatial locations $x_i (i = 1,2,...n)$, $Z(x_i)$ are the sample values at $x_i$. At an unknown target point $x_0$, the estimated value $\hat{Z}(x_0)$ can be expressed as a weighted average of the known observations $Z(x_i)$ (Wackernagel, 2013):

$$\hat{Z}(x_0) = \sum_{i=1}^{n} \lambda_i Z(x_i), \tag{12}$$

where $\lambda_i$ are the weighting coefficients of the known sample values $Z(x_i)$, which depend on the spatial autocorrelation structure of the sample values and should minimize the prediction error variance. Assuming the variable value $Z(x)$ can be modeled as a combination of a deterministic trend $\mu(x)$ and an auto-correlated random error $\varepsilon(x)$, $Z(x) = \mu(x) + \varepsilon(x)$, then the best linear unbiased prediction requires $E[\hat{Z}(x_0) - Z(x_0)] = 0$ and $Var[\hat{Z}(x_0) - Z(x_0)]$ is minimized. *Ordinary Kriging (OK)* assumes that the trend is constant but unknown, $\mu(x) = m$, while in *universal Kriging* (UK), the trend is assumed to be

a linear combination of some known covariables $f_l$, $\mu(x) = \sum_{l=1}^{k} \beta_l f_l$. *Universal Kriging* (UK) takes into account the relationship between the target variable and the auxiliary covariables. Soil, elevation, temperature, and remote sensing images were commonly used auxiliary covariables (Haberlandt, 1998; Li et al., 2014; McKenzie and Ryan, 1999; Semenov and Brooks, 1999).

     Both OK and UK were adopted to interpolate the CLIGEN input parameters in this study. Stepwise regression was

conducted to select appropriate covariables for UK. The longitude, latitude, elevation, and annual rainfall amount were found correlated with twelve groups of parameters CLIGEN with the exception of the SKEW P (Table 1) and were selected as auxiliary covariables for these twelve groups of parameters. SKEW P had low correlations with all four of these covariates



but good correlation with parameters MEAN P and SDEV P. Therefore, MEAN P and SDEV P were selected as covariables during the interpolation of SKEW P.

**2.4 Assessment of interpolation accuracy**

A leave-one-out cross-validation method was applied to evaluate the interpolation accuracy of OK and UK. The input parameters prepared using observation were denoted as $P_{ij}^{obs}$ ($i = 1, 2, …, 2405$ stations; $j = 1, 2, … 131$ input parameter values), and the corresponding inputs interpolated using OK (UK) as $P_{ij}^{OK}$ ($P_{ij}^{UK}$). For a specific parameter value $j_{th}$, assumed the value for the $i_{th}$ station was unknown and removed $P_{ij}^{obs}$ from all stations. Use the remaining stations to predict $P_{ij}^{OK}$

($P_{ij}^{UK}$) of $x_i$ using OK (UK), respectively. Following this procedure, two sets of input parameters for 2405 stations predicted by OK and UK were obtained and compared with parameters determined from observations to evaluate two interpolation methods.

Four indicators, *Nash-Sutcliffe efficiency coefficient* (NSE), *percent bias* (PBIAS), *root mean square error* (RMSE), and *RMSE-observations standard deviation ratio* (RSR), were selected to evaluate and compare the performances of OK and

UK as follows (Yin et al., 2019):

$$\text{NSE} = 1 - \frac{\sum_{i=1}^{n}(P_{ij}^{obs} - P_{ij}^{K})^2}{\sum_{i=1}^{n}(P_{ij}^{obs} - \overline{P^{obs}})^2}, \tag{13}$$

$$\text{PBIAS} = \frac{\sum_{i=1}^{n}(P_{ij}^{obs} - P_{ij}^{K})}{\sum_{i=1}^{n} P_{ij}^{obs}} * 100, \tag{14}$$

$$\text{RMSE} = \sqrt{\frac{1}{n}\sum_{i=1}^{n}(P_{ij}^{obs} - P_{ij}^{K})^2}, \tag{15}$$

$$\text{RSR} = \frac{RMSE}{\sqrt{\frac{1}{n}\sum_{i=1}^{n}(P_{ij}^{obs} - \bar{O})^2}} = \frac{\sqrt{\frac{1}{n}\sum_{i=1}^{n}(P_{ij}^{obs} - P_{ij}^{K})^2}}{\sqrt{\frac{1}{n}\sum_{i=1}^{n}(P_{ij}^{obs} - \overline{P^{obs}})^2}}. \tag{16}$$

By calculating of the above four indicators for each input parameter values, the better of the two interpolation techniques, OK and UK, was determined and applied to calculate the regionalization of CLIGEN input parameters for mainland China. A two-dimensional grid database was established at a spatial resolution of 10 km × 10 km based on 156 parameter layers in total.

Input parameters based on observed data and interpolated data using the better interpolation technique were input into

CLIGEN to evaluate the influence of regionalized parameters on the simulation. For each station, 100 years of continuous climate series were generated using the default CLIGEN stochastic seed without interpolation between months, and the simulated data predicted by $P^{obs}$ and $P^{K}$ were denoted as $G^{obs}$ and $G^{K}$, respectively. The maximum and minimum temperature, daily rainfall amount, storm duration, $i_p$ and $t_p$ of each simulation day were derived from $G_i^{obs}$ and $G_i^{k}$ for each station, and the maximum 30-min intensity ($I_{30}$) was calculated based on an assumed bi-exponential storm pattern (Yu, 2002).

*Absolute error* (AE) and *mean absolute errors* (MAE) were calculated to examine the differences between the two sets of





statistics for generated temperatures. *Relative error* (RE) and *mean absolute relative errors* (MARE) were calculated to examine the differences between the two sets of statistics for generated daily solar radiation, daily precipitation and sub-daily storm pattern:

$$|\text{AE}_i| = |G_i^{obs} - G_i^{k}|, \tag{17}$$

$$\text{MAE} = \frac{1}{2405} \sum_{i=1}^{2405} |(G_i^{obs} - G_i^{k})|, \tag{18}$$

$$|\text{RE}_i| = 100\% \, (G_i^{obs} - G_i^{k})/G_i^{obs}, \tag{19}$$

$$\text{MARE} = \frac{100\%}{2405} \sum_{i=1}^{2405} |(G_i^{obs} - G_i^{k})/G_i^{obs}|. \tag{20}$$

## 3 Results

### 3.1 Spatial-temporal distribution of CLIGEN input parameters

Thirteen groups of CLIGEN temperature and precipitation parameters from 2405 stations and solar radiation parameters from 130 stations were plotted to exhibit the inter-annual variation and the differences among parameters (Fig. 2). The average max-temperature and min-temperature, TMAX AV and TMIN AV (in unit of °F, 1°F = 1℃/1.8 + 32), and the average and standard deviation of solar radiation, SOL.RAD and SD SOL (in unit of Langley, 1 Ly = $4.184 * 10^{-2} MJ/m^2$) showed strong seasonality and the value became convergent from cold season to warm (Fig. 2a, 2c, 2e-f). The spatial distribution of CLIGEN temperatures and solar radiation related inputs in August based on the UK-interpolated results were depicted as examples (Fig. 3), from which we can find a differentiation rule for latitude and vertical zonality for TMAX AV, TMIN AV (Fig. 3a-b). SD TMAX and SD TMIN varied with season with a similar pattern and with generally higher values in spring and autumn (Fig. 3c-d), because these two seasons are transitional periods between warm and cold seasons when temperature fluctuation are larger.

**Fig. 2.**

**Fig. 3.**

The average and standard deviation of daily precipitation, MEAN P, S DEV P (in unit of inch, 1 inch = 25.4 mm), and the average monthly maximum 30-min intensity, MX5P (in unit of inch/h, 1 inch/h = 25.4 mm/h), showed a similar seasonal pattern with the parameter values becoming gradually higher from the cold season to the warm (Fig. 2g-h). Precipitation in China is influenced by the East Asian summer monsoon and the location relative to land and sea. From the spatial distribution of daily precipitation in August we found a general decreasing trend from southeast to southwest (Fig. 4a-b). The August rain belt is located in North and Northeast China, while the South China region is controlled by the subtropical high-pressure belt and experiences a summer drought. Therefore, MEAN P and MX.5P in North China was apparently greater than in South China. In comparison, skewness of daily precipitation, SKEW P, showed imperceptible differences among months and no apparent latitudinal or longitudinal zonality (Fig. 4c). This may be one of the reasons leading to the low spatial interpolation accuracy of SKEW P.





**Fig. 4.**

The wet-following-dry transition probability P(W/D) showed a clear inter-annual variability in that the probability increased from cold season to warm (Fig. 2j), while the wet-following-wet transition probability P(W/W) was characterized

by greater regional differences but smaller monthly variability for most stations compared with P(W/D) (Fig. 2k). The spatial-temporal variation in these two transition probabilities revealed the stepwise northward progress of East Asian monsoon and the North-South advance of the Frontal cyclone (Liao et al., 2004). Due to the pre-monsoon rainy season before June, strong convection in summer, and the retreating monsoon rain belt after August, the southern region was characterized by a longer rainy season than North China (Yu and Zhou, 2007). Therefore, P(W/W) of the southern region

was generally higher than other regions and its seasonal variations were relatively insignificant (Fig. 5b).

**Fig. 5.**

MX.5P of nine example stations showed the regional differences more clearly in that the parameters of southern stations were relatively higher (Fig. 5c). Differences among southern and northern stations became gradually smaller in the warm season. It should be noted that the narrower range of MX.5P in winter was partially related to the limited availability

of hourly data. Due to the restriction of low temperatures on siphon rain gauge observations, MX.5P in cold seasons were available for fewer stations than in warm seasons.

TimePk consists of 12 discrete values representing the cumulative distribution of time to peak intensity ranging from 0 to 1 for a specific location. The sixth value for TimePk represents the cumulative ratio of storms with peak intensity occurring before 1/2 duration, and related ratios for 2405 stations ranging from 60% to 80% (Fig. 2m). TimePk for nine

example stations shows the cumulative ratio of time to peak intensity in different regions, consistently indicating that most storms tend to occur earlier during the storms, with no obvious regional differences found for this parameter (Fig. 5d).

### 3.2 Evaluation of interpolated parameters using OK and UK

### 3.2.1 Parameters at a daily scale

The leave-one-out cross-validation showed that four groups of temperature parameters, TMAX AV, SD TMAX, TMIN

AV, SD TMIN, and four groups of precipitation parameters at daily scale, MEAN P, S DEV P, P(W/D) and P(W/W), were well predicted by *ordinary Kriging* (OK) and u*niversal Kriging* (UK). The average NSE over 12 months was greater than 0.88 for all these 8 groups of parameters. The PBIAS were all smaller than 1%, suggesting that parameters based on observation and interpolation have a very close average trend and showed no obvious bias. In contrast, the interpolated accuracy of two groups of solar radiation parameters, SOL.RAD, SD SOL, and the skewness coefficient of daily

precipitation, SKEW P, were not very satisfactory (Table 3), with NSE being 0.46-0.80 using OK and 0.66-0.85 using UK. The relatively lower interpolation accuracy of solar radiation related parameters was partially related to the sparsity of stations involved in the interpolation. Parameters related to daily average (TMAX AV, TMIN AV, SOL.RAD and MEAN P) were generally better predicted than corresponding parameters related to standard deviation (SD TMAX, SD TMIN, SD SOL





and S DEV P), and the skewness coefficient was the least accurately simulated. In addition, the interpolation accuracy

tended to be lower in the warm season (May to Sept.) compared with the yearly rest period (Fig. 6a-f).

**Table 3.**

In comparison with OK, the overall and monthly predicted accuracy using UK with auxiliary covariables obviously improved TMAX AV and TMIN AV in the warm season, SOL.RAD in the cold season and SD SOL in March. The predicted quality for SD TMAX, MEAN P, S DEV P, P(W|W), and P(W|D) was somewhat improved by UK, as these groups

of parameters already had high accuracy when using OK to interpolate, resulting in a small range of improvement. The predicted accuracy for the minimum temperature (SD TMIN) using the two techniques showed no evident difference, except for July, when the NSE of UK was obviously lower than OK and the reason was unclear. Although the prediction of SKEW P using UK was not as good as other parameters at a daily scale, the improvement compared with OK was quite obvious, as the average NSE over 12 months increased from 0.458 for OK to 0.769 for UK (Table 3). Predicted inputs using OK and UK

versus inputs based on observations from August were plotted to show the difference between two methods as examples (Fig. 7a-7k).

**Fig. 6.**

### 3.2.2 Parameters at a sub-daily scale

Cross-validation results showed that the interpolation accuracies of two storm pattern related parameters, MX.5P and

TimePk were not as good as precipitation related parameters on a daily scale. Four cross-validation statistics for these two parameters using two methods were numerically close (Table 3) for both parameters. After taking auxiliary covariates for interpolation using UK, the prediction improved only slightly. The annual variance of NSE based on OK and UK varied in a similar pattern within the year (Fig. 6l-m). For the parameter of TimePk, NSE of OK were slightly higher than that from UK from Jane to May, but reversed during the rest period. In comparison, MX.5P performed better than TimePk. The

interpolation accuracy of TimePk was the lowest among all 13 groups of input parameters (Table 3).

**Fig. 7.**

Interpolation accuracy has been adequately estimated through cross-validation, and these results agreed that the accuracy of interpolation results based on UK was generally higher than OK. Therefore, two sets of CLIGEN-simulated climate series using observed inputs and UK-interpolated inputs were generated and compared to further evaluate the

regionalized parameters using UK for simulation of CLIGEN.

### 3.3 Assessment of parameters' regionalization on the CLIGEN

### 3.3.1 Simulated climate elements at a daily scale

CLIGEN simulated daily temperature and solar radiation based on UK-interpolated input parameters agreed well with those simulated based on observed parameters. The average, standard deviation and skewness coefficient of generated daily



maximum temperature, minimum temperature, solar radiation and daily precipitation generated using observed and interpolated input parameters were calculated for each station, and the simulated accuracy of the average and standard deviation were found be better than that of the skewness coefficient. The NSE of the average and standard deviation were all greater than 0.97 for generated climate elements at a daily scale (Table 4). The NSE of the skewness coefficient for temperature and solar radiation ranged from 0.94-0.95, slightly lower than corresponding average and standard deviation. By

contrast, the NSE of the skewness coefficient of daily precipitation was as low as 0.48 (Table 5). This may be attributed to the lower interpolation accuracy of SKEW P, with the lowest accuracy among all input parameters (Table 3).

**Table 4.**

The *absolute error* (AE) of the average, standard deviation and skewness coefficient between the simulated daily temperature of $G^{obs}$ and $G^{UK}$ were statistically similar (Table 4). The *mean absolute errors* (MAEs) over 2405 stations were

all lower than 0.51℃. For daily solar radiation, the *relative errors* (REs) for three statistics were all lower than 2%, and the *mean absolute relative error* (MARE) were lower than 0.1%.

**Table 5.**

For generated daily precipitation, 94.1% and 91.4% of stations yielded REs of the average and standard deviation below 10%, and the MARE for 2405 stations were 3.72 and 4.56, respectively. Bias between annual rainy days of $G^{UK}$ and $G^{obs}$

was small as well. REs of 92.9% of stations were lesser than 10%. The frequency distribution of daily precipitation generated using two sets of inputs were well matched for most stations. Fig. 8a depicted the frequency distributions of simulated daily precipitation for Fuzhou station as an example, with RE slightly higher than MARE over 2405 stations. Meanwhile, some stations do not satisfactorily simulate the frequency distribution. The frequency distribution of Tuokexun, whose simulation quality was approximately the worst among 2405 stations was offered as an example (Fig. 8d). It showed

that the frequency of daily precipitation ranging from 0-1 mm was under-estimated, whereas that for values greater than 1 mm was over-estimated (Fig. 8d).

**Fig. 8.**

### 3.3.2 Simulated storm pattern related variables

The average and standard deviation of storm duration and the maximum 30-min intensity ($I_{30}$) generated using observed

and UK-interpolated input parameters possessed a generally small bias. The NSE of the average and standard deviation for both duration and $I_{30}$ were above 0.87. Compared with the average and standard deviation, the accuracy of skewness was the worst, with the NSE being 0.26 for the duration and 0.66 for the peak intensity index. Comparison of the frequency distribution of the duration and $I_{30}$ for Fuzhou station showed that the frequency of simulated storm patterns were well preserved using data employing UK-interpolated parameters (Fig. 8b-c). The frequency distribution of the duration and $I_{30}$

for Tuokexun station showed that interpolated parameters seemed to underestimate low values and overestimate high values (Fig. 8e-f).



## 4 Discussion

Both AE and RE indexes were adopted to evaluate the simulated results in this study. The RE index was applied for precipitation related outputs, while the AE index was applied for assessment of temperature-related outputs. This is because

we find that RE was not an appropriate index to evaluate temperature for some stations located in high latitude or high altitude areas where the mean annual temperature may be close to zero resulting in an extremely high derived RE. For example, the mean maximum temperature of Qian'an station (Fig. 1) using observed inputs was -0.01 ℃ and that using interpolated inputs was -0.33 ℃, resulting in an RE between the two values was 2912.7%, which was an extremely large error. However, the mean maximum temperature simulated using two data sets were very similar, with an AE of 0.32 ℃.

We've checked more than 100 stations with extremely high REs for maximum temperature, and all were in similar situation (Fig. 9). If RE was used to evaluate the simulated temperature, the actual simulation quality may be strongly underestimated. Therefore, AE were used to demonstrate errors between generated temperature based on observed and interpolated inputs.

**Fig. 9.**

The frequency distributions of CLIGEN simulated daily precipitation, duration and peak intensity at Tuokexun

station using observed inputs were all not well preserved by those simulated using UK-interpolated inputs (Fig. 8). The simulation quality for Tuokexun was almost the worst among 2405 stations, as REs for all these three precipitation related variables were greater than 99% of stations. This may be explained partially because Tuokexun is located in the northwest arid area of China (Fig. 1), with a station density of $0.97/10^4 \cdot km^2$, much lower than in the Eastern Monsoon Area (Table 7). Stations involved in the interpolation were separated by far distances, with a negative influence on the interpolation accuracy

(Oliver and Webster, 2014). Other stations with extremely low simulated quality similar to Tuokexun are almost located in the northwest arid area or Qinghai-Tibet Plateau where the station density is lower. The MAE for generated temperature and the MARE for generated precipitation related variables in the eastern monsoon area were the lowest among three physical-geographical regions of China (Table 7).

**Table 7.**

The number and density of weather stations for solar radiation were considerably less than for those for temperature and precipitation (Table 7). However, simulated daily solar radiation using the UK-interpolated parameters was in good agreement with that simulated using observation-based parameter values (Table 4). MARE for solar radiation across all stations was the lowest among all simulated weather elements. MAREs were similar for the three geographical regions with the difference among them varying from 0.08% to 0.13%. Solar radiation is characterized with much lower spatial variability

in comparison to that for temperature and precipitation. As a result, solar radiation-related parameters were easier to regionalize and parameter values could readily be interpolated for regions without limited observations.

**Fig. 10.**

CLIGEN-input parameters in the US is regionalized from 2600 stations using the *inverse distance weighted method* (IDW), which was employed in the initial attempt to regionalize CLIGEN input parameters. In this study, UK was adopted to



interpolate CLIGEN parameters for mainland China. Interpolated parameter values using IDW and UK were compared for four selected parameters in August as shown in Fig. 10. It can be seen that UK performed better than IDW for all four parameters selected. UK-interpolated parameter values were concentrated mostly along the 1:1 line. The NSEs of all four groups of parameters interpolated using UK were larger than those predicted using IDW. Noticeable improvement was noted for SKEW P, with the NSE improved from 0.27 to 0.74 using UK instead of IDW. Therefore, UK appears to be consistently

superior to IDW when regionalizing CLIGEN input parameters based on the limited comparison for selected parameters.

## 5 Data availability

   The girded CLIGEN input parameter dataset of China at 10km resolution is availability at the homepage of Climate Change impact assessment group – at http://clicia.bnu.edu.cn/data/cligen.html. Additional materials including the data manual and grid information are also availability at the same website and can be downloaded.

## 6 Conclusion

   The widely used stochastic weather generator CLIGEN can simulate long-term climate data to drive hydrological, soil erosion, and crop-yield models. Limitations in high spatial-temporal observations, especially at the sub-daily scale, have partially restricted its application. Daily temperature, daily precipitation, and hourly precipitation data for 2405 stations and daily solar radiation for 130 stations distributed across mainland China were collected to establish the CLIGEN input

parameter files and to explore an appropriate method for regionalizing these parameters from stations to the entire region. The predicted quality using two interpolation techniques, OK and UK, were compared and fully assessed, yielding the following results:

   1) UK generally performed better than OK when interpolating CLIGEN parameters. Compared with OK the interpolation accuracy was markedly improved for parameters TMAX AV, TMIN AV, SOL.RAD, SD SOL, SKEW P,

P(W/D) and P(W/W), and slightly improved for parameters SD TMAX, MEAN P and S DEV P. The comparative interpolation accuracies were numerically approximate between the two techniques.

   2) UK can accurately predict the temperature, solar radiation and precipitation input parameters for CLIGEN. The *Nash-Sutcliffe efficiency coefficient* (NSE) obtained using the observed parameters and UK-predicted parameters were all greater than 0.85 for most parameters expect for SD SOL, SKWE P and Time Pk. The interpolation accuracies for these final

three parameters were relatively lower, with NSEs greater than 0.66.

   3) Basic statistics and frequency distributions for CLIGEN-simulated climate elements using UK-interpolated parameters agreed well with those simulated using observations. The *mean absolute errors* (MAE) for the average, standard deviation and skewness coefficient for the two simulated series of temperature across 2405 stations were less than 0.51℃. The *mean absolute relative errors* (MAREs) for same statistics for simulated solar radiation were less than 0.1%. MAREs

for the average and standard deviation for precipitation, duration and $I_{30}$ are less than 5%, while errors for skewness coefficient for these three groups of parameters were less than 10.1%.

The developed gridded input parameter database can be applied using CLIGEN, with an established and reliable simulation quality, to the stochastic simulation of temperature, solar radiation and precipitation at a daily scale and to precipitation at a sub-daily scale for any single point in China. CLIGEN can simulate the dew point and wind as well, not

regionalized in this study. As a site-based weather generator, simulated climate series using CLIGEN are independent of each other and lack spatial correlations among stations. Further research might focus on the rebuilding of correlations among climate elements and between nearby sites.

**Competing Interests**

The authors declare that they have no conflict of interest.

**Acknowledgments**

This work was supported by the National Natural Science Foundation of China (No. 41877068) and China Postdoctoral Science Foundation (No. 2020M680433). We also would like to thank the high-performance computing support from the Center for Geodata and Analysis, Faculty of Geographical Science, Beijing Normal University [https://gda.bnu.edu.cn/].

**Author Contributions**

Wenting Wang calculated input parameters, developed the programming code, and wrote the original draft; Shuiqing Yin provided the main conceptualization, supervised the project, and reviewed the draft; Bofu Yu gave advises about the methodology and reviewed the draft; Shaodong Wang reviewed the draft.

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

**Table 1: Summary of CLIGEN input parameters and the data used for the calculation of parameters.**

| Inputs | Parameter description | Units | Number of parameters | Data used |
|---|---|---|---|---|
| TMAX AV | Average of daily maximum temperature | °F | Monthly, 12 in total | Daily temperature |
| SD TMAX | Standard deviation of daily maximum temperature | °F | Monthly, 12 in total | Daily temperature |
| TMIN AV | Average of daily minimum temperature | °F | Monthly, 12 in total | Daily temperature |
| SD TMIN | Standard deviation of daily minimum temperature | °F | Monthly, 12 in total | Daily temperature |
| SOL.RAD | Average of daily solar radiation | Langley | Monthly, 12 in total | Daily solar radiation |
| SD SOL | Standard deviation of daily solar radiation | Langley | Monthly, 12 in total | Daily solar radiation |
| MEAN P | Mean precipitation on rainy days | inch | Monthly, 12 in total | Daily precipitation |
| S DEV P | Standard deviation of precipitation on rainy days | inch | Monthly, 12 in total | Daily precipitation |
| SKEW P | The skewness coefficient of precipitation on rainy days | inch | Monthly, 12 in total | Daily precipitation |
| P(W/D) | The probability to wet day from dry day | | Monthly, 12 in total | Daily precipitation |
| P(W/W) | The probability to wet day | | Monthly, 12 in total | Daily |



| | | | | |
|---|---|---|---|---|
| | from wet day | | | precipitation |
| MX.5P | Maximum rainfall intensity per 30 min (0.5 hour) of a month | inch/h | Monthly, 12 in total | Hourly precipitation |
| TimePk | Relative time to the peak rainfall intensity | | Cumulative frequency, 12 in total | Hourly precipitation |


**Table 2: Data lengths for daily temperature, daily solar ration, daily and hourly precipitation from stations used in this study.**

| Data length (years) | Daily Temperature (1951-2014) | Daily rainfall (1951-2015) | Hourly rainfall (1951-2012) | Daily solar radiation (1957-2017) |
|---|---|---|---|---|
| <=10 | 19 | 16 | 215 | 5 |
| 10~20 | 17 | 19 | 34 | 9 |
| 20~30 | 20 | 20 | 94 | 44 |
| 30~50 | 269 | 240 | 1302 | 16 |
| >50 | 2080 | 2110 | 760 | 56 |
| Sum | 2405 | 2405 | 2405 | 130 |






**Table 3: Comparison of the accuracy of OK and UK using leave-one-out cross-validation.**

| Input parameters | NSE | | PBIAS (%) | | RMSE | | RSR | |
|---|---|---|---|---|---|---|---|---|
| | OK | UK | OK | UK | OK | UK | OK | UK |
| TMAX AV | 0.88 | 0.98 | -0.03 | -0.06 | 2.89 | 1.35 | 0.33 | 0.15 |
| SD TMAX | 0.92 | 0.93 | 0.00 | -0.01 | 0.36 | 0.35 | 0.28 | 0.27 |
| TMIN AV | 0.95 | 0.98 | -0.03 | 0.01 | 2.67 | 1.59 | 0.23 | 0.13 |
| SD TMIN | 0.90 | 0.89 | 0.02 | 0.08 | 0.45 | 0.46 | 0.32 | 0.33 |
| SOL.RAD | 0.80 | 0.85 | 0.28 | 0.25 | 30.85 | 25.89 | 0.45 | 0.38 |
| SD SOL | 0.60 | 0.66 | -0.08 | -0.10 | 14.87 | 13.66 | 0.62 | 0.57 |
| MEAN P | 0.94 | 0.95 | 0.00 | 0.16 | 0.03 | 0.02 | 0.25 | 0.22 |
| S DEV P | 0.94 | 0.94 | -0.08 | 0.03 | 0.05 | 0.05 | 0.25 | 0.24 |
| SKEW P | 0.46 | 0.77 | 0.08 | 0.09 | 0.73 | 0.48 | 0.73 | 0.48 |
| P(W/D) | 0.92 | 0.94 | 0.01 | -0.09 | 0.03 | 0.02 | 0.29 | 0.24 |
| P(W/W) | 0.91 | 0.94 | 0.02 | -0.01 | 0.04 | 0.03 | 0.31 | 0.24 |
| MX.5P | 0.88 | 0.88 | -0.06 | 0.08 | 0.11 | 0.11 | 0.34 | 0.34 |
| TimePk | 0.67 | 0.68 | 0.00 | 0.02 | 0.01 | 0.01 | 0.57 | 0.56 |




**Table 4: Comparison of daily temperature based on observation input parameters and UK interpolation parameters simulation.**

| Statistics | Daily maximum temperature | | | Daily maximum temperature | | | | Daily solar radiation (Ly) | | |
|---|---|---|---|---|---|---|---|---|---|---|
| | AV[1] | S DEV[2] | SKEW[3] | AV | S DEV | SKEW | | AV | S DEV | SKEW |
| NSE | 0.98 | 0.99 | 0.95 | 0.99 | 0.98 | 0.94 | | 0.99 | 0.98 | 0.94 |
| PBIAS | -0.1 | 0.05 | -0.33 | 0.01 | 0.05 | -0.23 | | 0.01 | 0.05 | -0.23 |
| RMSE | 0.68 | 0.25 | 0.03 | 0.79 | 0.35 | 0.04 | | 0.79 | 0.35 | 0.04 |
| RSR | 0.14 | 0.1 | 0.22 | 0.12 | 0.14 | 0.25 | | 0.12 | 0.14 | 0.25 |
| \|AE\|[4] | (%) | (%) | (%) | (%) | (%) | (%) | \|RE\| | (%) | (%) | (%) |
| < 1℃ | 93.7 | 99 | 100 | 86.2 | 97.5 | 100 | < 1% | 99.2 | 99.2 | 99.2 |
| < 2℃ | 98.5 | 99.8 | 100 | 97.4 | 99.6 | 100 | < 2% | 100 | 100 | 100 |
| < 5℃ | 99.8 | 100 | 100 | 99.9 | 100 | 100 | | | | |
| MAE(℃) | 0.51 | 0.21 | 0.02 | 0.34 | 0.14 | 0.02 | MARE(%) | 0.08 | 0.05 | 0.09 |

[1]The average, and [2]the standard deviation of daily maximum/minimum temperature simulated by CLIGEN.






**Table 5: Comparison of daily rainfall and yearly rain days based on observation input parameters and UK interpolation parameters simulation.**

| Estimation indicators | Daily precipitation | | | Annual rainy days | Storm duration | | | $I_{30}$ | | |
|---|---|---|---|---|---|---|---|---|---|---|
| | AV | S DEV | SKEW | AV | AV | S DEV | SKEW | AV | S DEV | SKEW |
| NSE | 0.98 | 0.97 | 0.48 | 0.97 | 0.92 | 0.87 | 0.26 | 0.99 | 0.98 | 0.66 |
| PBIAS | -0.06 | 0.27 | 0.94 | -0.01 | 0.28 | 0.73 | 0.13 | -0.34 | -0.2 | -0.15 |
| RMSE | 0.36 | 0.71 | 0.63 | 7.62 | 0.21 | 0.17 | 0.23 | 0.28 | 0.52 | 0.24 |
| RSR | 0.15 | 0.16 | 0.72 | 0.18 | 0.28 | 0.36 | 0.86 | 0.11 | 0.12 | 0.58 |
| \|RE\| | (%) | (%) | (%) | (%) | | | | | | |
| < 10% | 94.1 | 91.4 | 61.2 | 92.9 | 94.7 | 90.8 | 74.1 | 97.7 | 96.7 | 88.6 |
| < 20% | 98.6 | 98.6 | 87.4 | 98.4 | 98.8 | 97.9 | 93.5 | 99.7 | 99.4 | 98.3 |
| < 50% | 100 | 99.9 | 99.6 | 99.7 | 99.9 | 99.8 | 99.7 | 100 | 99.9 | 100 |
| MARE(%) | 3.72 | 4.56 | 10.07 | 4.09 | 3.47 | 4.61 | 7.71 | 2.36 | 3.07 | 5.08 |





**Table 7: Station density and simulation quality of three Chinese physical-geographical regions.**

|  | Eastern Monsoon Area | Northwest Arid Area | Qinghai-Tibet Plateau |
|---|---|---|---|
| *Temperature and precipitation* | | | |
| No. of stations | 2044 | 233 | 128 |
| Density(n/$10^4 \cdot km^2$) | 4.57 | 0.97 | 0.50 |
| MAE of Min Temperature (℃) | 0.44 | 0.90 | 0.93 |
| MAE of Max Temperature (℃) | 0.30 | 0.42 | 0.82 |
| MARE of Daily precipitation (%) | 3.13 | 6.92 | 7.25 |
| MARE of Duration (%) | 2.95 | 5.93 | 7.31 |
| MARE of $I_{30}$ (%) | 2.00 | 4.50 | 4.11 |
| | | | |
| *Solar radiation* | | | |
| No. of stations | 92 | 26 | 12 |
| Density(n/$10^4 \cdot km^2$) | 0.21 | 0.11 | 0.05 |
| MARE of Daily solar radiation (%) | 0.08 | 0.07 | 0.13 |



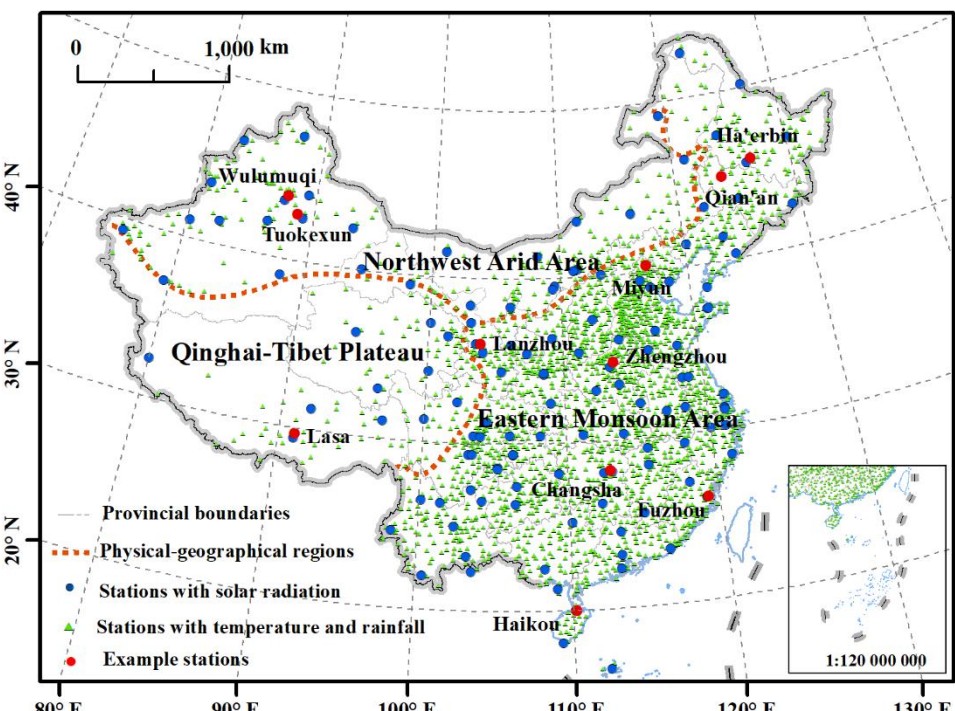

**Figure 1: Locations of meteorological stations used in this study.**



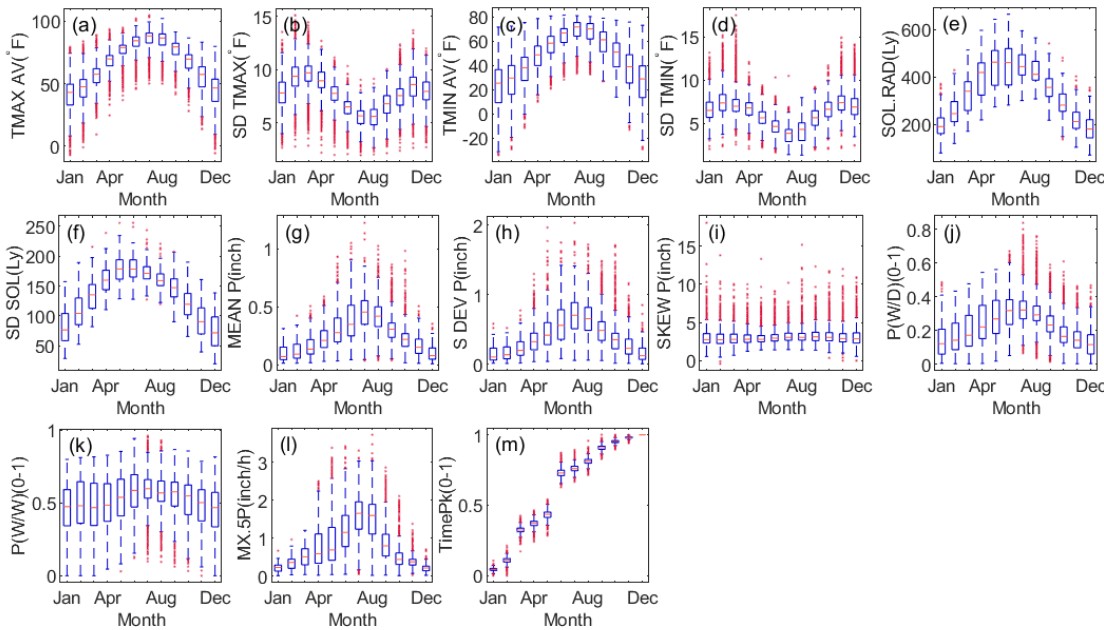

**Figure 2: Boxplot of CLIGEN temperature, solar radiation, and precipitation parameters obtained from observations in mainland China.**






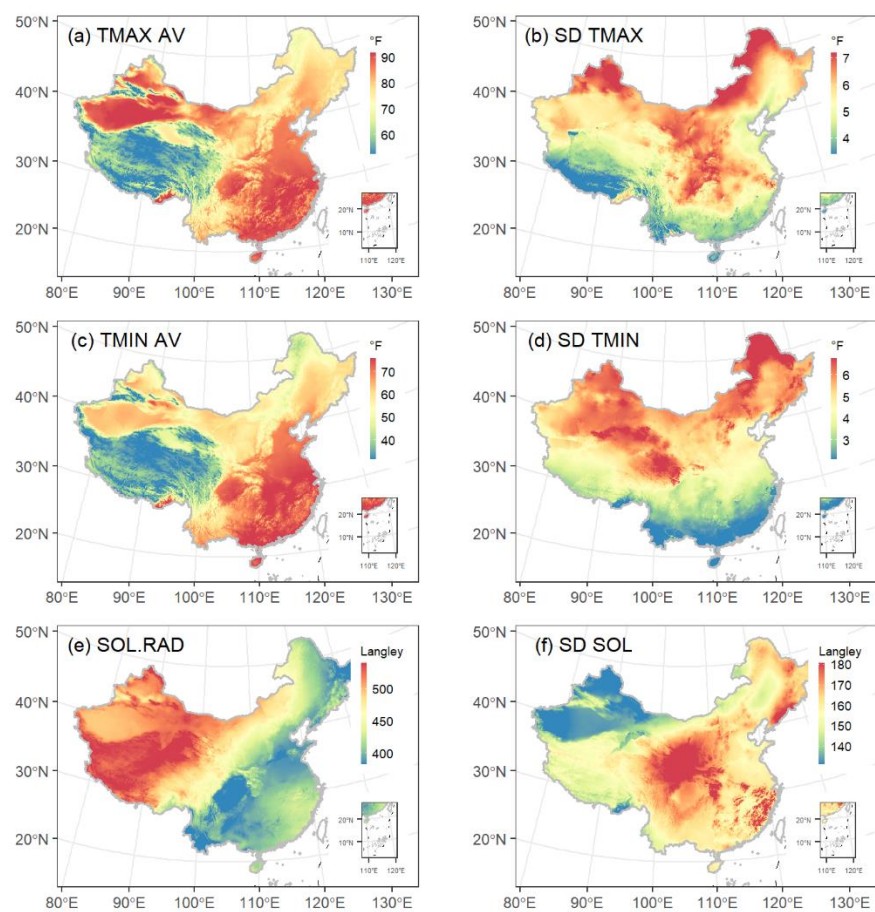

**Figure 3: Spatial distribution of CLIGEN temperature-related parameters of mainland China in August. All parameters were regionalized using *universal Kriging*.**




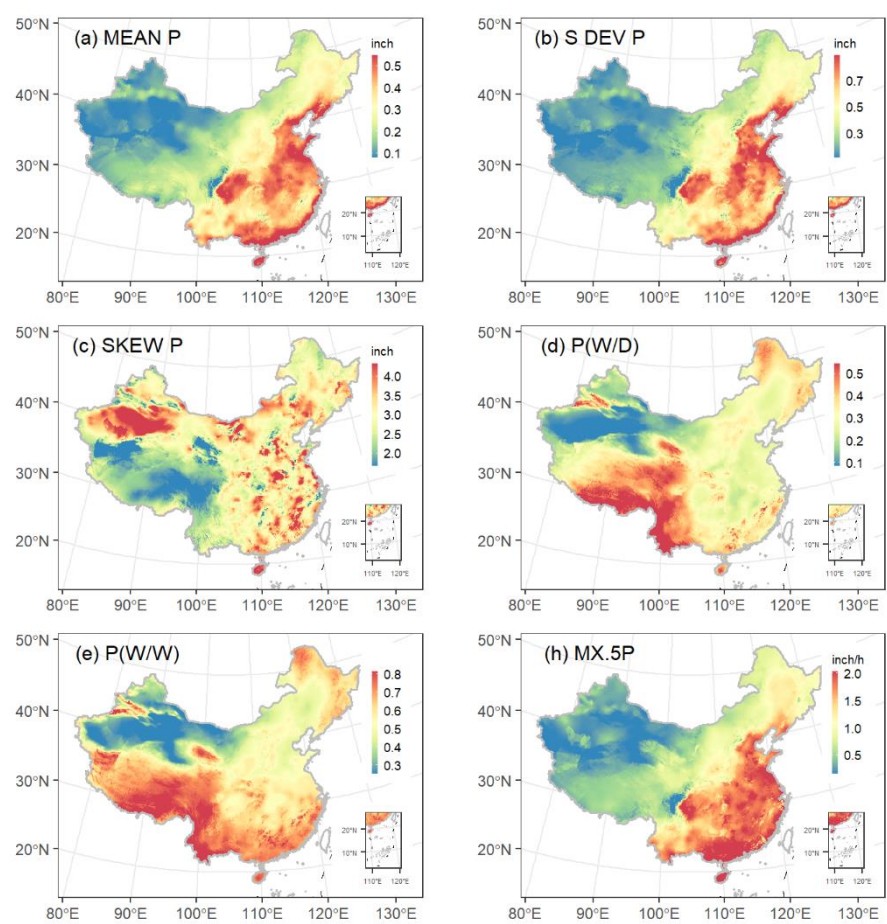

**Figure 4: Spatial distribution of CLIGEN precipitation related parameters of mainland China in August. All**

**parameters were regionalized using *universal Kriging*.**





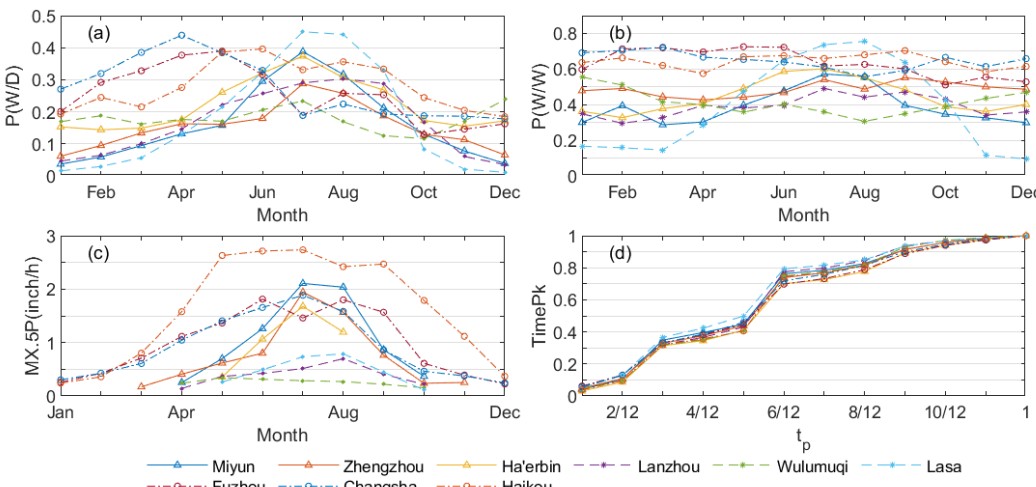

**Figure 5**: **P(W/D), P(W/W), MX.5P and TimePk of nine stations determined by observed daily precipitation.**




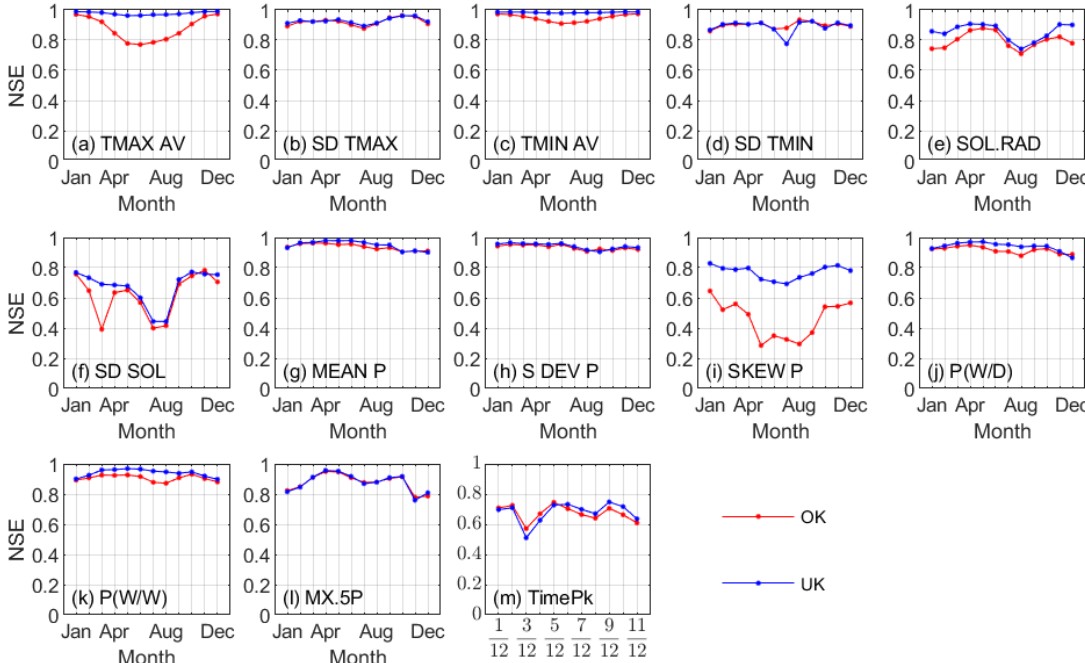

**Figure 6: Comparison of the interpolation quality in terms of the Nash-Stucliffe coefficient of efficiency (NSE) using** *ordinary Kriging* **(OK) and** *universal Kriging* **(UK) for temperature, solar radiation, and precipitation parameters.**




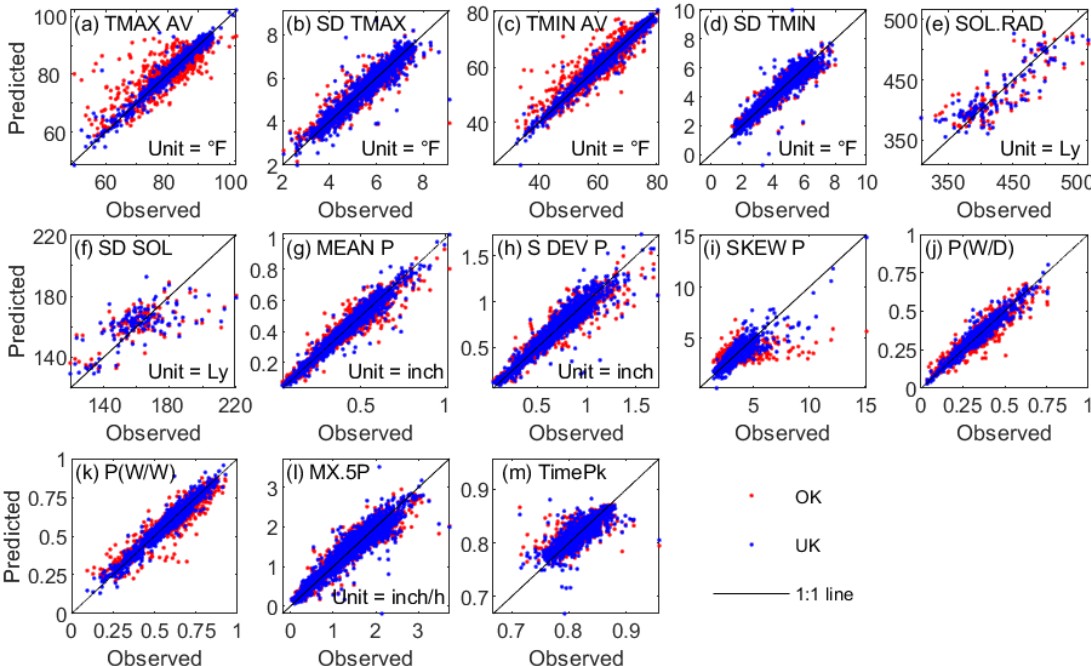

**Figure 7: Comparison of the interpolation quality using *ordinary Kriging* (OK) *and universal Kriging* (UK) for CLIGEN temperature, solar radiation, and precipitation parameters.**






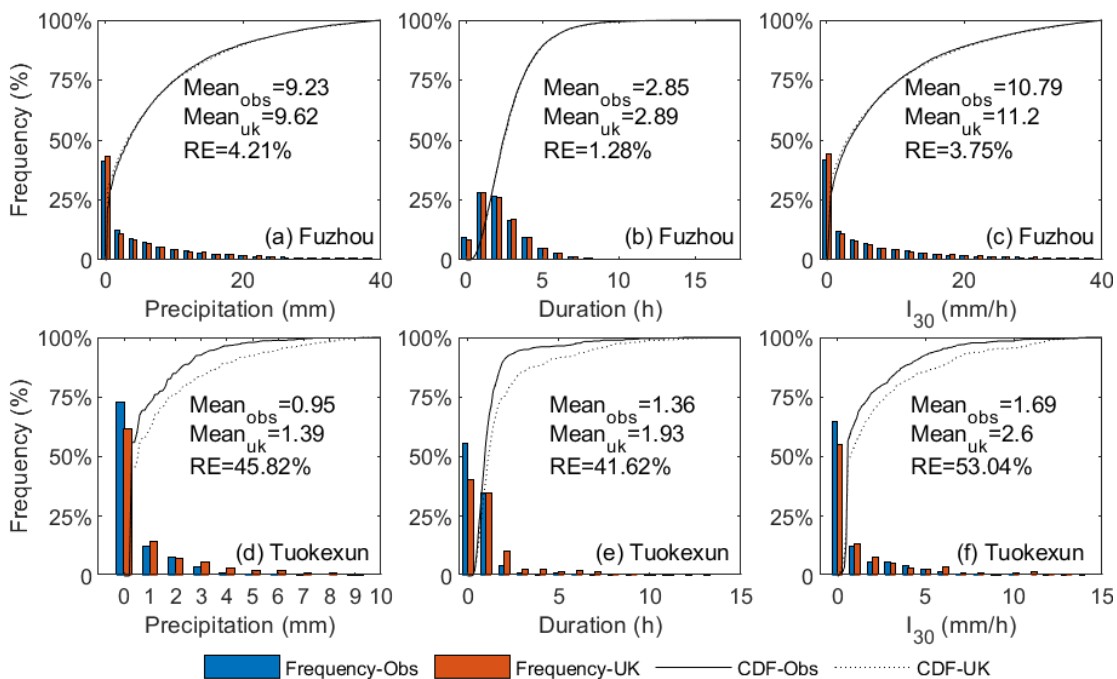

**Figure 8. Frequency distribution of daily precipitation, duration, and maximum 30-min intensity ($I_{30}$) generated by CLIGEN using inputs based on observations and interpolation predicted parameters: Fuzhou station (a-c) and Tuokexun station (d-f) as examples.**




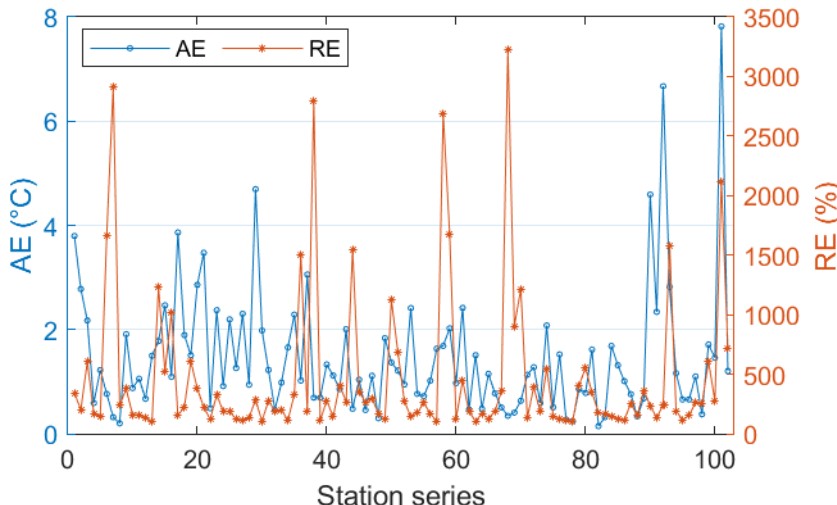

**Figure 9. Comparison of the absolute error (AE, ℃) and relative error (RE, %) of the simulated average of maximum temperature based on observed and UK-interpolated inputs by CLIGEN for 102 stations with extremely large RE.**






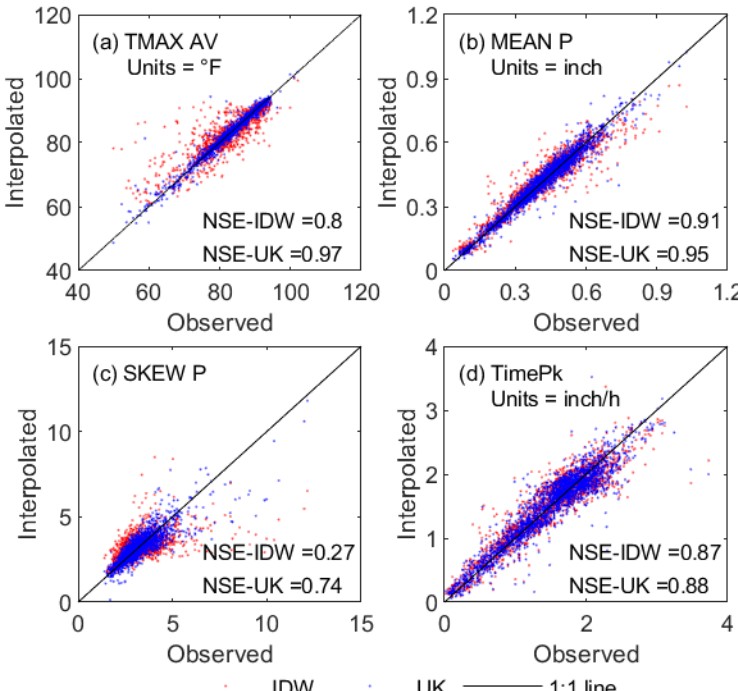

**Figure 10**: **Comparison of interpolation quality using** *universal Kriging* **(UK) and the** *inverse distance weighted method* **(IDW) for CLIGEN temperature and precipitation parameters for 2405 sites in summer (August).**
