# Peer review of "CLIGEN Parameter Regionalization for Mainland China"

_Earth System Science Data, 2020_

## Author Comment (AC1)

**Reviewer #1**

This paper presents a valuable dataset that may encourage greater application of soil erosion modelling in China. The results of the cross-validation show good agreement, and an interesting comparison of ordinary and universal kriging is made. Some details of the methodology could be clarified, particularly in terms how the covariates and parameter layers are used in universal kriging. Importantly, spatial interpolation error could also be discussed more. Overall, the paper is well written and describes a valuable dataset.

Comment (1): Line 22: Word "in" is not needed in this context.

Response: It will be revised.

Comment (2): Line 23: typo: available

Response: It will be corrected.

Comment (3): Line 78: WGEN has not been defined. This is another stochastic weather generator?

Response: WGEN is the abbreviation of the Weather GENerator, which is a weather generator developed by Richardson and Wright (1984). The description and reference will be added to the manuscript. (Reference: Richardson, C.W., Wright, D.A.: WGEN: A model for generating daily weather variables, 1984.)

Comment (4): Line 152: This statement could be clarified: "Rainfall intensity is basically assumed to be ranked from high to low in CLIGEN".

Response: In CLIGEN (Nicks et al., 1995), also Arnold and Williams (Arnold and Williams, 1989; Williams et al. 1984), it is assumed that the magnitude of precipitation intensity decreases exponentially from the maximum rate when time distribution of precipitation intensities is discarded. We will revise it to make it clearer.

Comment (5): Line 173: How was TimePk determined in Wang et al. (2018b). More information would be helpful beyond the fact there was available hourly precipitation and MX.5P values to estimate TimePk. More detail could be given about how the other intensity parameter, MX.5P was determined if it requires high-resolution data.

Response: For hourly data as we collected, the time to peak intensity, $t_p$, can be calculated for every storm directly using equations (9) to (11) listed in the manuscript. For example, if the storm duration is 4 hours, and the peak intensity occurs in the 2$^{nd}$ hour, then $t_p = (2\text{-}0.5)/4 = 0.375$ based on equation (10). Then using equation (11), parameters TimePk can be obtained for all stations. From Wang et al. (2018b), we know that two sets of TimePk parameter values prepared using hourly rainfall and 1-min rainfall generate very similar CLIGEN outputs. Therefore, we used hourly rainfall to prepare TimePk for all stations following equations (9) to (11) for this study.

Ideally, MX.5P values should be prepared using rainfall intensity data with an observed time interval of no more than 30 minutes. Depending on the temporal resolution, $I_{30}$ can be calculated directly from moving averages of the original data over successive 30-min.

More sentences will be added to explain how TimePk and MX.5P were calculated in the revised manuscript.

Comment (6): Line 191: typo: CLIGEN parameters.

Response: It will be revised.

Comment (7): Line 191: Clarify how twelve groups were used.

Response: Twelve groups means twelve months for each parameter, and the expression is not clear. The statement will be revised to "The longitude, latitude, elevation, and annual rainfall amount were found correlated with the parameters one for each month for CLIGEN" to make it clearer.

Comment (8): Line 195: How many iterations of leave-one-out cross-validation were there that produced the four performance metrics? Is this equal to the number of stations?

Response: Yes, this is correct. For each interpolated parameter, the leave-one-out cross-validation procedures were iterated for 2405 times, which equals to the number of stations. More descriptions will be added to the manuscript for clarification.

Comment (9): Line 197: How was the 131 input parameters number arrived at?

Response: There was a typo here. There were 13 groups of input parameters required for CLIGEN for temperature, precipitation and solar radiation, and in each group, there are 12 parameters. The total number of interpolated parameters is 155, which is equal to 12 parameters $\times$ 13 groups-1, as the 12$^{th}$ parameter of TimePk is always equal to 1. Therefore, there were 155 parameters involved in the interpolation, and this will be revised in the manuscript.

Comment (10): Line 210: Word "of" not needed.

Response: It will be revised.

Comment (11): Line 210: Clarify how the 156 number of parameter layers was arrived at.

Response: Please refer to the comment (9).

Comment (12): Line 234: Clarify "the value became convergent from cold season to warm".

Response: It will be revised to "spatial variance became smaller from the cold season to the warm one".

Comment (13): Line 285: Interpolation accuracy is stated here to be temporally dependent, but more discussion of how it is spatially dependent would be helpful. I would guess that in data sparse areas, interpolation error is much higher. The leave-one-out cross-validation does not account for the fact that data sparse areas could actually represent large parts of the total interpolated area, so it could be the case that error would be much higher if more observations

were available to check error in data sparse areas. Is it indeed the case that error is higher in western China? Would it be possible to make an error map for MEANP and TMAX as examples? Or, consider some way of presenting spatial error.

Response: We agree with you that the interpolation error for data sparse areas is higher. We've discussed the influence of spatial distribution of stations on the interpolation accuracy in the second paragraph in Discussion. We calculated and compared the mean absolute relative errors (MAREs) derived from leave-one-out cross-validation for three regions in China (the Eastern Monsoon Area, the Northwest Arid Area and Qinghai-Tibet Plateau) with different station density. Results were listed in Table 7 and showed that the station density has an influence on the quality of the interpolation. Error in the Eastern Monsoon Area is the lowest and the highest in the Qinghai-Tibet Plateau.

Considering that the station density is quite sparse in the western region of China, and the leave-one-out cross validation can't show the interpolation error in regions without stations. We've plotted the standard error of the interpolation results for two parameters, TMAX AV and MEAN P in August as an example (Fig. 1). It can be seen from the figures that the errors are relatively high in the western part, especially in the south-western part, where is a large area without stations and characterized with the highest standard errors for both parameters. The following figure will be added to the discussion part of the manuscript to make this clearer.

[Figure]

Fig. 1. Spatial distribution of the standard error for interpolation results of TMAX AV and

MEAN P in August using Universal Kriging.

Comment (13): Table 3: RMSE for TimePk of 0.01 is very small considering TimePk ranges

from 0-1. Why is NSE particularly low for TimePk? Is it the small numeric scale of TimePk?

Response: Thank you for your comment. We've checked the computation and data that the results are without mistake. TimePk is a group of dimensionless parameters, which represents the cumulative probability distribution of the time to peak. The equation for NSE is,

$$NSE = 1 - \frac{\sum_{i=1}^{n}(P_{ij}^{obs} - P_{ij}^{K})^2}{\sum_{i=1}^{n}(P_{ij}^{obs} - \overline{P^{obs}})^2}$$

where $i$ = 1, 2, …2045 stations, $j$ = 1, …11 parameters. It's only 11 parameters of TimePk need to be interpolated because the 12th parameter of TimePk is always equal to 1, which is known already. NSE for 11 parameters can be calculated, and the average value over 11 NSEs is the number listed in Table 3 in the manuscript.

As the equation for NSE represents the ratio between predicted errors and the variation of the observed values. The variation of the observed values depends on its specific shape of distribution. For each of the 11 TimePk values, the variation of observed values across the whole study area is relatively small compared with the predicted errors (as showed in Fig. 7m in the manuscript), leading to a particularly low NSE value. However, if we put all 11 parameters of TimePk together to calculate NSE, it will become much better and equal to 0.998 for both OK and UK.

Reviewer's comment reminds us that NSE may be not a good indicator for parameters in interval scales which are without true zero. Therefore, we will reorganize results listed in Table 3 that the NSEs will only be kept for parameters in ratio scales (MEAN P, S DEV P, SKEW P, SOL.RAD, and SD SOL). RMSE will be kept for all the 13 groups of parameters. The statements, figures and tables in Results will be reorganized correspondingly.

---

## Author Comment (AC2)

**Reviewer #2**

This research provides useful information and a database for application of the CLIGEN weather generator within China. Technically the results and paper are sound. A number of minor grammatical/spelling issues need to be addressed.

Specific comments on the manuscript are:

Comment (1): Line 32 - this states "global climate models (GCMs)" - however, the original meaning of "GCMs" is "General Circulation Models". Please carefully consider which should be mentioned here (or both).

Response: We agree with the reviewer that the original meaning of "GCMs" is "General Circulation Models". While, nowadays it often refers to "Global Climate Models (GCMs)" along with the development of Atmosphere, Ocean and Land coupled models. GCM in this manuscript is referred to as Global Climate Models.

Comment (2): Lines 39-40 - change "accompanied by the process-based soil erosion model Water Erosion Prediction Project (WEPP) by the United" to "accompanying the process-based Water Erosion Prediction Project (WEPP) model from the United".

Response: It will be revised.

Comment (3): Lines 49-50 - this states "if modeling wind-induced snow drift is not needed", however, wind is used in WEPP for other calculations, including ET estimation using the Penman equation, and snow melting calculations. Snow drift is not currently simulated by WEPP, and actually is not mentioned in Flanagan and Livingston (1995). Suggest this text be deleted.

Response: Thank you for pointed out this problem, and it has will be deleted.

Comment (4): Line 85 - change "have attempted" to "have been attempted".

Response: It will be revised.

Comment (5) Line 91 - change "in WEPP application" to "in the WEPP application".

Response: It will be revised.

Comment (6) Line 113 - change "have been quality" to "had been quality".

Response: It will be revised.

Comment (7) Line 121 - change "in the North China" to "in North China".

Response: It will be revised.

Comment (8) Line 133 - change "transition probabilities P(W|D) and P(W|W)" to "day transition probabilities P(W|W) and P(W|D)".

Response: It will be revised.

Comment (9): Line 141 - change "amount in CLIGEN" to "depths in CLIGEN".

Response: It will be revised.

Comment (10): Line 149 - change "MX.5P are" to "MX.5P values are".

Response: It will be revised.

Comment (11): Line 187 - change "images were commonly" to "images are commonly".

Response: It will be revised.

Comment (12): Line 191 - change "with twelve groups of parameters CLIGEN" to "with the twelve groups of parameters for CLIGEN".

Response: It will be revised.

Comment (13): Line 198 - change "assumed" to either "we assumed" or "it was assumed".

Response: It will be revised.

Comment (14): Line 199 - change "Use the" to "We then used the".

Response: It will be revised.

Comment (15): Line 201 - change "to evaluate two" to "to evaluate the two"

Response: It will be revised.

Comment (16): Lines 206-209 - punctuation marks (commas, period) after Equations 13-16 are probably not needed, and could be deleted.

Response: It will be revised and the punctuation marks will be deleted.

Comment (17): Line 210 - change "parameter values" to "parameter value".

Response: It will be revised.

Comment (18): Lines 224-227 - punctuation marks (commas, period) after Equations 17-20 are probably not needed, and could be deleted.

Response: It will be revised and the punctuation marks will be deleted.

Comment (19): Line 231 - change "to exhibit the" to "to examine the".

Response: It will be revised.

Comment (20): Line 239 - change "temperature fluctuation" to "temperature fluctuations".

Response: It will be revised.

Comment (21): Line 247 - change "North China was" to "North China were".

Response: It will be revised.

Comment (22): Line 256 - change "of East Asian" to "of the East Asian".

Response: It will be revised.

Comment (23): Line 271 - change "storms tend to" to "peak intensities tend to".

Response: It will be revised.

Comment (24): Line 276 - for the word "universal", italicize the entire word including the first letter "u".

Response: It will be revised.

Comment (25): Line 303 - change "higher than that from UK" to "higher than those from UK".

Response: It will be revised.

Comment (26): Line 304 - change "from Jane to May" to "from January to May".    Change "the rest period" to "the June to December period".

Response: It will be revised.

Comment (27): Line 307 - change "results agreed that" to "results indicated that".

Response: It will be revised.

Comment (28): Line 308 - change "higher than OK" to "than those based on OK".

Response: It will be revised.

Comment (29): Line 311 - change "on the CLIGEN" to "on the CLIGEN outputs".

Response: It will be revised.

Comment (30): Line 335 - change the word "under-estimated" to "underestimated".

Response: It will be revised.

Comment (31): Line 336 - change the word "over-estimated" to "overestimated".

Response: It will be revised.

Comment (32): Lines 352-353 - You could have used degrees Kelvin instead of degrees Centigrade. That would have resulted in better Relative Error calculations.

Response: Thank you for your suggestion. Convert the unit from Celsius to Kelvin will lead to lower relative errors for temperature. But temperature is parameter in interval scale and without true zero, for which the absolute error is a better indicator to describe the estimated error.

Response: It will be revised.

Comment (34): Line 354 - change "temperature simulated using two data sets" to "temperature values simulated using the two data sets".

Response: It will be revised.

Response: It will be revised.

Response: It will be revised.

Comment (37): Line 387 - change "The girded CLIGEN" to "The gridded CLIGEN". change "is availability at the hompage of" to "is available at the homepage of the".

Response: It will be revised.

Comment (38): Line 390 - consider changing "Conclusion" to "Summary and Conclusions".

Response: It will be revised.

Comment (39): Line 404 - change "parameters expect for" to "parameters except for".

Response: It will be revised.

Comment (40): Line 410 - change "are less than" to "were less than".

Response: It will be revised.

Comment (41): Line 427 - change "gave advises" to "provided advice".

Response: It will be revised.

Comment (42): Line 441 - change this citation to: "Flanagan, D.C., and Livingston, S.J. (eds.): WEPP User Summary. NSERL Report No. 11, USDA-Agricultural Research Service, National Soil Erosion Research Laboratory, West Lafayette, Indiana, USA, 1995.".

Response: It will be revised.

Comment (43): Line 443 - change "107-110, 2001" to "American Society of Agricultural Engineers, St. Joseph, Michigan, USA, pp. 107-110, 2001.".

Response: It will be revised.

Comment (44): Line 446 - change "1-9, 2014" to "American Society of Agricultural and Biological Engineers, St. Joseph, Michigan, USA, 8 pp., 2014.".

Response: It will be revised.

Comment (45): Lines 447-448 - This is not correct. RUSLE2 science documentation should be credited to USDA-Agricultural Research Service (2013). Full citation would be: "USDA-ARS: Science Documentation, Revised Universal Soil Loss Equation, Version

2 (RUSLE2), USDA-Agricultural Research Service, Washington, D.C., USA, 2013."   Also need to change reference to this in the text at all locations.

Response: It will be revised.

Comment (46): Line 449 and throughout all of the list of references - be sure to capitalize all words that are part of a journal name. Here it should be the Journal of Hydrologic Engineering.

Response: It will be revised.

Comment (47): Line 496 - do not capitalize words in the title of a journal article. So here, title should be "Seasonality and three-dimensional structure of interdecadal change in the East Asian monsoon".

Response: It will be revised.

Comment (48): Page 17, Table 1 - change "The probability to wet day from dry day" to "The probability of a wet day following a dry day" and change "The probability to wet day from wet day" to "The probability of a wet day following a wet day".

Response: It will be revised.

Comment (49): Page 29, Figure 7 - caption - do not italacize the word "and" in "and universal Kriging".

Response: It will be revised.

---

## Author Response (AR1)

Dear Editor and Reviewers:

We would like to thank the editor and reviewers for reviewing the paper and providing comments. Detailed responses to all reviewers' comments were provided below. We believe that answering to the reviewers' comments/suggestions substantially improved the manuscript.

**Reviewer #1**

This paper presents a valuable dataset that may encourage greater application of soil erosion modelling in China. The results of the cross-validation show good agreement, and an interesting comparison of ordinary and universal kriging is made. Some details of the methodology could be clarified, particularly in terms how the covariates and parameter layers are used in universal kriging. Importantly, spatial interpolation error could also be discussed more. Overall, the paper is well written and describes a valuable dataset.

Comment (1): Line 22: Word "in" is not needed in this context.

Response: It has been deleted, see line 23 of the revised manuscript.

Comment (2): Line 23: typo: available

Response: It has been corrected to available, see line 24 of the revised manuscript.

Comment (3): Line 78: WGEN has not been defined. This is another stochastic weather generator?

Response: WGEN is the abbreviation of the Weather GENerator, which is a weather generator developed by Richardson and Wright (1984). More description and the corresponding reference have been added to the manuscript, see line 76-78 of the revised manuscript.

Comment (4): Line 152: This statement could be clarified: "Rainfall intensity is basically assumed to be ranked from high to low in CLIGEN".

Response: In CLIGEN (Nicks et al., 1995), also Arnold and Williams (1989), it's basically assumed that the magnitude of precipitation intensity decreases exponentially from the

maximum rate when time distribution of precipitation intensities is discarded. We have added more description in line 151-152 of the revised manuscript to make it clearer.

Comment (5): Line 173: How was TimePk determined in Wang et al. (2018b). More information would be helpful beyond the fact there was available hourly precipitation and MX.5P values to estimate TimePk. More detail could be given about how the other intensity parameter, MX.5P was determined if it requires high-resolution data.

Response: For hourly data as we collected, the time to peak intensity, $t_p$, can be calculated for every storm directly using equations (9) to (11) listed in the revised manuscript. For example, if the storm duration is 4 hours, and the peak intensity occurs in the 2nd hour, then $t_p = (2\text{-}0.5)/4 = 0.375$ based on equation (10). Then using equation (11), parameters TimePk can be obtained for all stations. From Wang et al. (2018), we know that two sets of TimePk parameter values prepared using hourly rainfall and 1-min rainfall generate very similar CLIGEN outputs. Therefore, we used hourly rainfall to prepare TimePk for all stations following equations (9) to (11) for this study.

Ideally, MX.5P values should be prepared using rainfall intensity data with an observed time interval of no more than 30 minutes. Depending on the temporal resolution, $I_{30}$ can be calculated directly from moving averages of the original data over successive 30-min.

More sentences have been added to explain how TimePk and MX.5P were calculated in the revised manuscript, see line 147-150 and line 172-176.

Comment (6): Line 191: typo: CLIGEN parameters.

Response: It has been revised.

Comment (7): Line 191: Clarify how twelve groups were used.

Response: Twelve groups means twelve months for each parameter, and the expression is not clear. The statement has been revised to "The longitude, latitude, elevation, and annual rainfall

amount were found correlated with the parameters one for each month for CLIGEN" to make it clearer, see line 193-194.

Comment (8): Line 195: How many iterations of leave-one-out cross-validation were there that produced the four performance metrics? Is this equal to the number of stations?

Response: Yes, this is correct. For each interpolated parameter, the leave-one-out cross-validation procedures were iterated for 2405 times, which equals to the number of stations. More descriptions have been added to line 199-202 of the revised manuscript for clarification.

Comment (9): Line 197: How was the 131 input parameters number arrived at?

Response: There was a typo here. There were 13 groups of input parameters required for CLIGEN for temperature, precipitation and solar radiation, and in each group, there are 12 parameters. The total number of interpolated parameters is 155, which is equal to 12 parameters × 13 groups-1, as the 12$^{th}$ parameter of TimePk is always equal to 1. Therefore, there were 155 parameters involved in the interpolation, and this has been revised in line 201-203 of the revised manuscript.

Response: It has been revised, see line 212 of the revised manuscript.

Comment (11): Line 210: Clarify how the 156 number of parameter layers was arrived at.

Response: Please refer to the comment (9).

Comment (12): Line 234: Clarify "the value became convergent from cold season to warm".

Response: It has been revised to "spatial variance became smaller from the cold season to the warm one", see line 239 of the revised manuscript.

Comment (13): Line 285: Interpolation accuracy is stated here to be temporally dependent, but more discussion of how it is spatially dependent would be helpful. I would guess that in data

sparse areas, interpolation error is much higher. The leave-one-out cross-validation does not account for the fact that data sparse areas could actually represent large parts of the total interpolated area, so it could be the case that error would be much higher if more observations were available to check error in data sparse areas. Is it indeed the case that error is higher in western China? Would it be possible to make an error map for MEANP and TMAX as examples? Or, consider some way of presenting spatial error.

Response: We agree with you that the interpolation error for data sparse areas is higher. We've discussed the influence of spatial distribution of stations on the interpolation accuracy in the second paragraph in the second paragraph of the Discussion. We calculated and compared the mean absolute relative error (MARE) derived from leave-one-out cross-validation for three regions in China (the Eastern Monsoon Area, the Northwest Arid Area and Qinghai-Tibet Plateau) with different station density. Results were listed in Table 6 and showed that the station density has an influence on the quality of the interpolation. Error in the Eastern Monsoon Area is the lowest and the highest in the Qinghai-Tibet Plateau.

Considering that the station density is quite sparse in the western region of China, and the leave-one-out cross validation can't show the interpolation error in regions without stations. We've plotted the standard error of the interpolation results for two parameters, TMAX AV and MEAN P in August as an example as following. It can be seen from the figures that the errors are relatively high in the western part, especially in the south-western part, where is a large area without stations and characterized with the highest standard errors for both parameters. The following figure has been added to the discussion part as Fig. 9, and the statements has been revised to make this clearer, see Line 366-370 of the revised manuscript.

[Figure]

Fig. 1. Spatial distribution of the standard error for interpolation results of TMAX AV and MEAN P in August using Universal Kriging.

Comment (13): Table 3: RMSE for TimePk of 0.01 is very small considering TimePk ranges from 0-1. Why is NSE particularly low for TimePk? Is it the small numeric scale of TimePk?

Response: Thank you for your comment. We've checked the computation and data that the results are without mistake. TimePk is a group of dimensionless parameters, which represents the cumulative probability distribution of the time to peak. The equation for NSE is,

$$\text{NSE} = 1 - \frac{\sum_{i=1}^{n}(P_{ij}^{obs} - P_{ij}^{K})^2}{\sum_{i=1}^{n}(P_{ij}^{obs} - \overline{P^{obs}})^2}$$

where $i = 1, 2, …2045$ stations, $j = 1, …11$ parameters. It's only 11 parameters of TimePk need to be interpolated because the 12[th] parameter of TimePk is always equal to 1, which is known already. NSE for 11 parameters can be calculated, and the average value over 11 NSEs is the number listed in Table 3 of the original manuscript.

As the equation for NSE represents the ratio between predicted errors and the variation of the observed values. The variation of the observed values depends on its specific shape of distribution. For each of the 11 TimePk values, the variation of observed values across the whole study area is relatively small compared with the predicted errors (as showed in Fig. 7m in the manuscript), leading to a particularly low NSE value. However, if we put all 11 parameters of TimePk together to calculate NSE, it will become much better and equal to 0.998 for both OK and UK.

Reviewer's comment reminds us that NSE may be not a good indicator for parameters in

interval scales which are without true zero, neither for PBIAS. Therefore, we reorganized results listed in Table 3 that the NSE and PBIAS were only kept for parameters in ratio scales (MEAN P, S DEV P, SKEW P, SOL.RAD, and SD SOL). RMSE were kept for all the 13 groups of parameters. The statements in Results were reorganized correspondingly. Fig.6 were revised, the NSE of 12 months were replaced with RMSE.

**Reviewer #2**

This research provides useful information and a database for application of the CLIGEN weather generator within China. Technically the results and paper are sound. A number of minor grammatical/spelling issues need to be addressed.

Specific comments on the manuscript are:

Comment (1): Line 32 - this states "global climate models (GCMs)" - however, the original meaning of "GCMs" is "General Circulation Models". Please carefully consider which should be mentioned here (or both).

Response: We agree with the reviewer that the original meaning of "GCMs" is "General Circulation Models". While, nowadays it often refers to "Global Climate Models (GCMs)" along with the development of Atmosphere, Ocean and Land coupled models. GCM in this manuscript is referred to as Global Climate Models.

Comment (2): Lines 39-40 - change "accompanied by the process-based soil erosion model Water Erosion Prediction Project (WEPP) by the United" to "accompanying the process-based Water Erosion Prediction Project (WEPP) model from the United".

Response: It has been revised, see line 39-40 of the revised manuscript.

Comment (3): Lines 49-50 - this states "if modeling wind-induced snow drift is not needed", however, wind is used in WEPP for other calculations, including ET estimation using the Penman equation, and snow melting calculations. Snow drift is not currently simulated by

WEPP, and actually is not mentioned in Flanagan and Livingston (1995). Suggest this text be deleted.

Response: Thank you for pointing out this problem, and it has been deleted.

Comment (4): Line 85 - change "have attempted" to "have been attempted".

Response: It has been revised, see line 83 of the revised manuscript.

Comment (5) Line 91 - change "in WEPP application" to "in the WEPP application".

Response: It has been revised, see line 89 of the revised manuscript.

Comment (6) Line 113 - change "have been quality" to "had been quality".

Response: It has been revised, see line 111 of the revised manuscript.

Comment (7) Line 121 - change "in the North China" to "in North China".

Response: It has been revised, see line 119 of the revised manuscript.

Comment (8) Line 133 - change "transition probabilities $P(W|D)$ and $P(W|W)$" to "day transition probabilities $P(W|W)$ and $P(W|D)$".

Response: It has been revised, see line 131 of the revised manuscript.

Comment (9): Line 141 - change "amount in CLIGEN" to "depths in CLIGEN".

Response: It has been revised. See 139 of the revised manuscript.

Comment (10): Line 149 - change "MX.5P are" to "MX.5P values are".

Response: It has been revised. See line 147 of the revised manuscript.

Comment (11): Line 187 - change "images were commonly" to "images are commonly".

Response: It has been revised. See line 189 of the revised manuscript.

Comment (12): Line 191 - change "with twelve groups of parameters CLIGEN" to "with the twelve groups of parameters for CLIGEN".

Response: This sentence has been revised to "parameters one for each month for CLIGEN", see line 194 of the revised manuscript.

Comment (13): Line 198 - change "assumed" to either "we assumed" or "it was assumed".

Response: This paragraph has been revised and the sentence "assumed ..." has been deleted. See line 200-205 of the revised manuscript.

Comment (14): Line 199 - change "Use the" to "We then used the".

Response: This paragraph has been revised and the sentence "Use the..." has been deleted. See line 200-205 of the revised manuscript.

Comment (15): Line 201 - change "to evaluate two" to "to evaluate the two"

Response: This paragraph has been revised and the sentence "to evaluate two..." has been deleted. See line 200-205 of the revised manuscript.

Comment (16): Lines 206-209 - punctuation marks (commas, period) after Equations 13-16 are probably not needed, and could be deleted.

Response: It has been revised that the punctuation marks has been deleted from all equations.

Comment (17): Line 210 - change "parameter values" to "parameter value".

Response: This paragraph has been revised, and "for each input parameter values" has been deleted. See line 212 of the revised manuscript.

Comment (18): Lines 224-227 - punctuation marks (commas, period) after Equations 17-20 are probably not needed, and could be deleted.

Response: It has been revised and the punctuation marks has been deleted.

Comment (19): Line 231 - change "to exhibit the" to "to examine the".

Response: It has been revised, see line 231 of the revised manuscript.

Response: It has been revised, see line 244 of the revised manuscript.

Response: It has been revised, see line 253 of the revised manuscript.

Response: It has been revised, see line 261 of the revised manuscript.

Response: It has been revised, see line 275 of the revised manuscript.

Response: It has been revised, see line 281 of the revised manuscript.

Response: It has been revised, see line 305 of the revised manuscript.

Comment (26): Line 304 - change "from Jane to May" to "from January to May".   Change "the rest period" to "the June to December period".

Response: This sentence has been revised to "For TimePk, the RMSE using OK were slightly lower than those using UK for the $3^{th}$, $4^{th}$, and $5^{th}$ parameters, but slightly higher for the others".

TimePk is a group of dimensionless parameters, which represents the cumulative probability distribution of the time to peak and is not calculated by month.

Comment (27): Line 307 - change "results agreed that" to "results indicated that".

Response: It has been revised, see line 307 of the revised manuscript.

Comment (28): Line 308 - change "higher than OK" to "than those based on OK".

Response: It has been revised, see line 308 of the revised manuscript.

Comment (29): Line 311 - change "on the CLIGEN" to "on the CLIGEN outputs".

Response: It has been revised that the "outputs" has been added to line 311 of the revised manuscript.

Comment (30): Line 335 - change the word "under-estimated" to "underestimated".

Response: It has been revised, see line 346 of the revised manuscript.

Comment (31): Line 336 - change the word "over-estimated" to "overestimated".

Response: It has been revised, see line 346 of the revised manuscript.

Comment (32): Lines 352-353 - You could have used degrees Kelvin instead of degrees Centigrade. That would have resulted in better Relative Error calculations.

Response: Thank you for your suggestion. Convert the unit from Celsius to Kelvin will lead to lower relative errors for temperature. But temperature is parameter in interval scale and without true zero, for which the absolute error is a better indicator to describe the estimated error.

Comment (33): Line 353 - change "was 2912.7%" to "of 2912.7%".

Response: It has been revised, see line 354 of the revised manuscript.

Comment (34): Line 354 - change "temperature simulated using two data sets" to "temperature values simulated using the two data sets".

Response: It has been revised, see line 355 of the revised manuscript.

Comment (35): Line 378 - change "US is regionalized" to "US are regionalized".

Response: It has been revised, see line 379 of the revised manuscript.

Comment (36): Line 384 - change "NSE improved" to "NSE improving".

Response: It has been revised, see line 385 of the revised manuscript.

Comment (37): Line 387 - change "The girded CLIGEN" to "The gridded CLIGEN".   change "is availability at the hompage of" to "is available at the homepage of the".

Response: It has been revised, see line 389 of the revised manuscript.

Comment (38): Line 390 - consider changing "Conclusion" to "Summary and Conclusions".

Response: It has been revised, see line 392 of the revised manuscript.

Comment (39): Line 404 - change "parameters expect for" to "parameters except for".

Response: It has been revised, see line 405 of the revised manuscript.

Comment (40): Line 410 - change "are less than" to "were less than".

Response: It has been revised, see line 411 of the revised manuscript.

Comment (41): Line 427 - change "gave advises" to "provided advice".

Response: It has been revised, see line 428 of the revised manuscript.

Comment (42): Line 441 - change this citation to: "Flanagan, D.C., and Livingston, S.J. (eds.): WEPP User Summary. NSERL Report No. 11, USDA-Agricultural Research Service, National Soil Erosion Research Laboratory, West Lafayette, Indiana, USA, 1995.".

Response: This reference has been deleted.

Comment (43): Line 443 - change "107-110, 2001" to "American Society of Agricultural Engineers, St. Joseph, Michigan, USA, pp. 107-110, 2001.".

Response: It has been revised, see line 447-448 of the revised manuscript.

Comment (44): Line 446 - change "1-9, 2014" to "American Society of Agricultural and Biological Engineers, St. Joseph, Michigan, USA, 8 pp., 2014.".

Response: It has been revised, see line 452-453 of the revised manuscript.

Comment (45): Lines 447-448 - This is not correct. RUSLE2 science documentation should be credited to USDA-Agricultural Research Service (2013). Full citation would be: "USDA-ARS: Science Documentation, Revised Universal Soil Loss Equation, Version 2 (RUSLE2), USDA-Agricultural Research Service, Washington, D.C., USA, 2013." Also need to change reference to this in the text at all locations.

Response: The citation of this reference has been revised, see line 453-454 of the revised manuscript.

Comment (46): Line 449 and throughout all of the list of references - be sure to capitalize all words that are part of a journal name. Here it should be the Journal of Hydrologic Engineering.

Response: The references have been revised.

Comment (47): Line 496 - do not capitalize words in the title of a journal article. So here, title should be "Seasonality and three-dimensional structure of interdecadal change in the East Asian monsoon".

Response: It has been revised.

Comment (48): Page 17, Table 1 - change "The probability to wet day from dry day" to "The probability of a wet day following a dry day" and change "The probability to wet day from wet day" to "The probability of a wet day following a wet day".

Response: It has been revised.

Comment (49): Page 29, Figure 7 - caption - do not italacize the word "and" in "and universal Kriging".

Response: It has been revised.

In addition, during the revision, we found a data format error related to the solar radiation. The magnitude of the input parameters related to solar radiation are correct in the original datasets, but a decimal point is missing. For example, the following table shows the change of monthly parameters of SOL.RAD (the monthly average daily solar radiation in Langley) in Aihui station. During the simulation, the Fortran code of CLIGEN read this decimal point as one digit, which resulted in the magnitude of CLIGEN-generated solar radiation is 0.1 times smaller in the original version.

| Original SOL.RAD | 127 | 223 | 341 | 400 | 458 | 502 | 450 | 391 | 309 | 216 | 139 | 98 |
| Revised SOL.RAD | 127. | 223. | 341. | 400. | 458. | 502. | 450. | 391. | 309. | 216. | 139. | 98. |

We've rewrite the CLIGEN input parameters' files of all these 96676 interpolated files, and updated the dataset in the website (http://clicia.bnu.edu.cn/data/cligen.html). We also re-calculated and re-evaluated the CLIGEN-generated solar radiation, and revised the results listed in Table 4 and Table 6 and corresponding statements in the Results and Discussion, see line 316-321, line 326-327, line 372-377.

---

## Referee Report (RR1)

Good improvements.

Line 147: Typo

Line 178: Use term "kriging" instead of "kriging interpolation" so that the word "interpolation" is not redundantly used in this sentence.

Line 401: Typo

---

## Author Response (AR2)

**Reviewer #1**

Response: It has been revised to "Ideally, MX.5P values should be prepared using rainfall data with a resolution of 30 min or less."

Response: "interpolation" has been deleted.

Response: It has been revised to "For the remaining parameters, the comparative interpolation accuracies were numerically approximate between the two techniques."

**Reviewer #1**

Response: It has been revised to "Ideally, MX.5P values should be prepared using rainfall data with a resolution of 30 min or less."